# Ergonomic Design and Performance Evaluation of H-Suit for Human Walking

**DOI:** 10.3390/mi13060825

**Published:** 2022-05-25

**Authors:** Leiyu Zhang, Zhenxing Jiao, Yandong He, Peng Su

**Affiliations:** 1Beijing Key Laboratory of Advanced Manufacturing Technology, Beijing University of Technology, Beijing 100124, China; zhangleiyu@bjut.edu.cn (L.Z.); jiaozhenxing0109@163.com (Z.J.); heyandong13@163.com (Y.H.); 2School of Electromechanical Engineering, Beijing Information Science and Technology University, Beijing 100192, China

**Keywords:** soft robotic suit, hip assistance, gait prediction, performance evaluation, EMG signal

## Abstract

A soft exoskeleton for the hip flexion, named H-Suit, is developed to improve the walking endurance of lower limbs, delay muscle fatigue and reduce the activation level of hip flexors. Based on the kinematics and biomechanics of the hip joints, the ergonomic design of the H-Suit system is clearly presented and the prototype was developed. The profile of the auxiliary forces is planned in the auxiliary range where the forces start at the minimum hip angle, reach the maximum (120 N) and end at 90% of each gait cycle. The desired displacements of the traction unit which consist of the natural and elastic displacements of the steel cables are obtained by the experimental method. An assistance strategy is proposed to track the profile of the auxiliary forces by dynamically adjusting the compensation displacement *L*_c_ and the hold time *Δt*. The influences of the variables *L*_c_ and *Δt* on the natural gaits and auxiliary forces have been revealed and analyzed. The real profile of the auxiliary forces can be obtained and is consistent with the theoretical one by the proposed assistance strategy. The H-Suit without the drive unit has little effect on the EMG signal of the lower limbs. In the powered condition, the H-Suit can delay the muscle fatigue of the lower limbs. The average rectified value (ARV) slope decreases and the median frequency (MNF) slope increases significantly. Wearing the H-Suit resulted in a significant reduction of the vastus lateralis effort, averaged over subjects and walking speeds, of 13.3 ± 2.1% (*p* = 2 × 10^−5^).

## 1. Introduction

The expanding elderly and disabled population poses considerable challenges to the current healthcare system [1]. A lower-limb exoskeleton has been developed to reduce the metabolic consumption of human walking or running [2,3,4,5] and is widely used in rehabilitation and strength enhancement [6]. According to the stiffness of the robot’s main components, the lower-limb exoskeleton can be divided into two types, rigid and soft exoskeletons. The traditional rigid exoskeleton robots, which are worn by lower limbs and moved with the human body, imitate the biological structure of human lower extremities. The rigid linkages or components are taken as the main mechanical backbone [7,8]. The rigid exoskeleton can provide gait correction for patients with dyskinesia and walking assistance for healthy persons (such as BLEEX [9], HAL System [10], EKSO [11], and ReWalk [12]). However, the additional mass of lower extremities and the axis misalignment of the human–machine system after wearing can easily lead to the increase of additional torques, a disorder of natural gait, and dislocation of auxiliary forces [13,14,15,16]. The energy consumption of the human body will rise substantially and it is difficult to be applied to humans fast walking or running. Currently, assistive devices focus more on comfort and personalization [17], in order to overcome the shortcomings of the rigid exoskeleton mentioned above, many scholars have studied soft exoskeletons with better wearing comfort (SEU-EXO [14], Exosuit [18], and Myosuit [19]), namely the soft power-assisted system, which can assist the lower limbs in the state of walking or running. The flexible driving units, such as the Bowden cable, pneumatic muscle and airbag, are applied to transmit the auxiliary forces/moments to the hip, knee or ankle joints [20].

The soft power-assisted system supported by the lower limb skeleton transfers the auxiliary forces through the flexible elements (nylon belt, flexible cable, pneumatic muscles, etc.) distributed along the skin surface to the single or multiple joints [21], and reduces the additional loads and metabolic consumption of the corresponding flexor/extensor muscles. Ding et al. [22] developed a functional coat called Exosuit which was worn on the skin surface of lower limbs. The multi-joint assistance was provided for the hip extension and ankle plantar flexion through Bowden cables and an external traction unit. Kim and Ding et al. [18,23] designed an Exosuit with a waist belt and tight trousers as the substrate and developed a portable drive device worn on the waist. The Exosuit system only provided auxiliary forces for hip extension and was suitable for walking and running at a specific speed. Aiming at the gravity intensive activities (sitting, standing, squatting, etc.) of the knee joint, Schmidt et al. [19] developed Myosuit, which consisted of corsetry, ligament layer and active layer, to achieve the function of the external bionic muscle by controlling the contraction and relaxation of the active layer. The pneumatic muscle was applied in a Power Assist Wear [24] to improve the independent walking ability of the elderly and the tension force was transmitted to the hip and knee joints by nylon belts. Tian et al. [25] developed a unilateral power-assisted exoskeleton based on the pneumatic muscle, which could assist the flexion and extension of the hip/knee joints and reduce the muscle activation by 14.6%. A flexible ankle-assisted exoskeleton developed by John et al. [26] proposed a cross-line power suit with four Bowden cables. The cables were distributed along the front and rear sides of the thigh and could be controlled independently to provide auxiliary force/moment for the hip joint. Based on the above studies, a soft assistance suit system, which consists of the suit body, Bowden cables and traction unit, is proposed for the hip flexion.

Based on the kinematics and biomechanics of the hip joint, a new soft power-assisted system named H-Suit is proposed to provide a suitable auxiliary force for the forward flexion of the hip joint. An assistance strategy is established to produce a desired profile of the auxiliary force. The fatigue and activation of hip flexors (iliopsoas muscle, rectus femoris and vastus lateralis, etc.) are analyzed comprehensively under different experimental conditions and walking speeds. Understanding how the H-Suit affects the hip flexors of human bodies is fundamental for quantifying its benefits and drawbacks, guiding a continuous data-driven design refinement. Furthermore, the H-Suit system can be worn for a long time for healthy persons, especially soldiers who need to walk a long distance with a heavy load. Besides, the additional mass and gait interference of lower limbs can be reduced as much as possible with the help of the ergonomic design method.

## 2. Ergonomic Design of the H-Suit

### 2.1. Kinematics and Biomechanics of Hip Joint

The lower extremities of humans are mainly composed of the hip, knee and ankle joints. The main activities of the lower limbs (walking or running) are mainly composed of flexion/extension (FL/EX) of the three joints. Hence, the muscle groups that dominate hip FL/EX are analyzed in this paper. During human walking, the flexor and extensor muscles work together to control FL/EX movements. Hip flexors mainly consist of the iliopsoas muscle, rectus femoris and vastus lateralis [27]. The main action interval of hip flexors and extensors is complementary in a single gait cycle *T* [28]. The activation times of the flexors are concentrated in the middle stage of gait cycle *T*, namely the late support phase and the prometaphase of the swing phase. The corresponding activation of extensors is distributed at the beginning and end stage of a gait cycle. The corresponding activation intensity can be easily detected and measured [29,30]. Besides, the leg lift that occurred in the flexion movements should overcome the gravity of lower limbs rapidly and have greater energy expenditures. Hence, the flexion movements are selected as the assistance object in this study. The kinematics and biomechanics of the hip joint in a single gait cycle *T* are shown in Figure 1 where *θ*_h_ and *T*_h_ denote the rotation angle and biological torque of the hip joint, respectively.

### 2.2. Ergonomic Design of H-Suit

The H-Suit consists of a soft exosuit, external traction units and Bowden cables. The soft exosuit is composed of waist belts, thigh belts, inertial measurement units (IMUs) and load cells, as shown in Figure 2a. The waist and thigh belts are embraced around the waist and thighs separately. The anchors used to fix Bowden cables are sutured on the belt surfaces. In order to improve the rated output of the traction unit and the smoothness of auxiliary torques, the optimal positions of anchors (Figure 2b) were selected in our previous study [31]. Three IMUs are respectively stuck on the front sides of the waist and thigh to detect the kinematics information of the hip joint. Besides, the load cells are connected in series between the steel cable and the thigh anchor to measure the auxiliary forces in real-time. The power of the external traction units is transmitted to the soft exosuit through Bowden cables which consist of sheaths and steel cables. The H-Suit has two identical traction units for the right and left legs. One end of the sheath is fixed at the waist anchor and the other at the frame of the traction units. Similarly, one end of the steel cable is installed at the thigh anchor and the other at the slider of traction units. Based on the gait information, an expected profile of auxiliary forces can be applied to the H-Suit and assist flexions movements by controlling the traction displacements.

According to the average body sizes of our research group (height 175 cm, weight 75 kg), four bottom garments are selected and bought. The waist and thigh belts consist of open frames and Velcro. The open frames are designed and fitted with the physiological structures of the human body to improve the corresponding twisting resistance (Figure 2c) [32]. Four short nylon belts (width 25 mm) are sewn on the anchor locations denoted by *A*, *B*, *C* and *D* (Figure 2a). Two types of anchor components with butterfly shapes are fixed on the short belts. The sheath is inserted and bonded in the hole of the upper component. The steel cable with a plug is mounted in the lower component, passed through the upper one and enters the sheath. In order to increase the cable resilience during the no-assistance period, a compression spring is added between the upper and lower components. Two load cells are installed at the upper short belts in series to measure the auxiliary forces. Furthermore, shoulder straps are connected at the front and back of the waist belt to avoid the dislocation of the H-Suit. In order to collect the gait information of lower limbs, two IMUs are fixed on the left and right thigh belts, respectively, and a third one at the waist as reference. IMUs only collect the motion angle in the human sagittal plane.

There are two external traction units for the right and left lower extremities. The synchronous belt with advantages of smooth motion and small backlash is adopted in the traction unit. A slider is reciprocally pulled by a synchronous belt driven by a brushless motor (TBM7646, Kollmorgen Inc., New York, NY, USA) and the steel cable is connected to this slider (Figure 2d). The tension and release of the cable are accomplished by those reciprocal motions. The traction displacements of the cable are measured by a draw-wire displacement sensor (WFS1000; Fiaye Inc., Shanghai, China).

The total mass except for the traction unit is 0.995 kg where 0.8 kg is distributed around the waist and 0.195 kg around the thigh. The H-Suit has few constraints on the dexterity of low limbs and daily activities. When the subjects wear the H-Suit, they can easily complete the actions of half-kneeling, squatting down and bending over, etc. (Figure 2e). The weight of each traction unit is 1.5 kg. The traction units are installed on a cart and used to test the assistance strategy. The cart can move with the subjects when evaluating the performance of the H-Suit.

### 2.3. Control Unit

The control unit of H-Suit is mainly composed of a PC, an interactive interface, and the control board of MCU. The gait detection/prediction algorithm and the control algorithm are integrated into the software. PC can communicate with MCU through a serial port to realize data receiving and sending, and all data of the gait, auxiliary forces and traction displacement are displayed constantly in the interactive interface. MCU receives the control instructions from PC and controls the brushless motors through the CAN bus to realize the assistance of hip flexion. The data of IMUs, load cells and displacement sensors are collected by MCU and sent to the PC in real-time. Besides, a DC power supply, electric relays and limit switches are necessary to guarantee the stable work of the control unit. In order to serve the performance evaluation of H-Suit expediently, a cart with four floors is designed and manufactured. The two traction units connected to the exosuit body through Bowden cables are fixed on the second floor and the control unit on the third floor. The exosuit is folded up and placed on the bottom layer when not in use. The subjects can wear the H-Suit and walk on the treadmill or indoors where the cart is put aside.

## 3. Assistance Strategy of H-Suit

It is necessary to analyze the activations of hip flexors and the characteristics of the lower limb’s gait and optimize the assistance strategy. According to the parameters of lower limbs and the stiffness of the exosuit body, the desired auxiliary forces are transformed into the trajectory of the traction unit. Then, the assistance strategy of H-Suit is established to ensure smooth assistance for lower limbs.

### 3.1. Auxiliary Force

The assistance performance is directly affected by the starting/ending time and the peak value of the auxiliary forces. The hip flexion which dominates 40% of the whole gait cycle only can occur when going from the maximum extension (minimum *θ*_min_ of hip angle *θ*) to the maximum flexion (maximum *θ*_max_). Further, the hip flexion is divided into two stages: ① The first stage is from *θ*_min_ to the zero-point *θ*_zero_, ② The second one is from *θ*_zero_ to *θ*_max_, as shown in Figure 3a. Compared with the first stage, the second one processes a longer duration. Besides, the zero-point *θ*_zero_, the beginning of the second stage, can be easier predicted through the prediction algorithm of “Newton + triple exponential smoothing” proposed in the previous research [31]. When the lower limb reaches the maximum extension position (Figure 3a), the flexors are activated and start isometric contraction to produce biological forces for the hip flexion. When the hip angle *θ* is about zero in the region of flexion, the activation of the flexor muscles reaches the maximum and produces the maximum biological forces for hip flexion [33]. The activation decreases to a stable level when the lower limb reaches 90% of a gait cycle *T*. Therefore, the profile of the auxiliary force *F*_t_ begins at *t* = *t_θ_*_,min_, reaches the peak value *F*_t,max_ at *t* = *t_θ_*_,zero_, and ends at *t* = *t*_90%_, as shown in Figure 3b.

Based on the biological force of the flexor muscles [27], the 15% of the biological peak force is taken as *F*_t, max_ which is equal to 120 N. The profile of *F*_t_ is designed and the corresponding expression of *F*_t_ is obtained as follows:(1)$$F_{t}=109.5\sin\left[\left(1103P-1.298\right)\times \frac{\pi }{180}\right]+21.13\cos\left[\left(2826P+10.83\right)\times \frac{\pi }{180}\right]$$
where *P* is the percentage of the gait cycle *T* (60% < *P* < 90%). The time *t_θ_*_,min_ and *t_θ_*_,zero_ can be substituted by the parameter *P* which stays in dynamic changes according to the time *t_θ_*_,min_ and gait cycle *T*. When the angle *θ* reaches *θ*_min_, the percentage *P* corresponding to the time *t_θ_*_,min_ is obtained.

### 3.2. Desired Displacement of the Traction Unit

Based on the profile of forces *F*_t_, the desired displacement of the traction unit can be calculated which is closely related to the hip angle *θ* and the total stiffness *k*_total_ of the H-Suit. A coordinate system *xoy* is built at the center *o* of the hip joint and the axis *ox* is horizontal, as shown in Figure 4a. The vectors ***oA*** and ***oB*** are the position vector of the waist and thigh anchors. According to the optimal position of *A* and *B*, the length of ***oA*** and ***oB*** denoted *l_oA_* and *l_oB_* can be calculated. The corresponding included angle *α* indicates the dynamic position of the thigh during the walking. The changes in the length *l_AB_* of ***AB***, without the auxiliary forces, are the natural traction displacement *L*_n_ to adapt the positions of the thigh. The initial length *l*_0_ of ***AB*** at the standing posture can be obtained by the cosine theorem:(2)$$l_{0}=\sqrt{l_{oA}^{2}+l_{oB}^{2}-2l_{oA}\cdot l_{oB}\cdot \cos\alpha_{0}}$$
where α_0_ is the angle α in the standing posture.

Assuming that the angle of the anchor *A* relative to the hip center is constant during the human walking, the anchor *B* is rotated around the center *o* and the angle *α* is obtained:(3)$$\alpha =\alpha_{0}-\theta$$Hence, the length *l_AB_* of ***AB*** can be calculated:(4)$$l_{AB}=\sqrt{l_{oA}^{2}+l_{oB}^{2}-2l_{oA}\cdot l_{oB}\cdot \cos\alpha }$$

The natural displacement *L*_n_ of the traction unit can be acquired:(5)$$L_{n}=l_{AB}-l_{0}$$

The auxiliary force *F*_t_ is applied to the anchors along the vector ***AB***. There exist elastic deformations in the exosuit, Bowden cables and local tissues. Due to the intercoupling of the H-Suit and local tissues, it is difficult to measure and calculate the stiffness of three parts denoted by *k*_s_, *k*_c_ and *k*_t_, respectively. However, the total stiffness *k*_total_ can be measured by the experimental method [22,23]. A subject wore H-Suit and the right lower limb flexed forwards 15°, as shown in Figure 4b. A traction force *F*_t,ex_ was applied to the anchors from 0 to 120 N with the help of the traction unit. The subject kept still to complete the experiments. The displacements of the traction unit were measured to indicate the elastic deformations *L*_e_ by the displacement sensor. The external force *F*_t,ex_ at the equal intervals of displacements are presented in Figure 4c.

According to the measured data, the expression of *k*_total_ is fitted and calculated where *k*_total_ = 2100 N/m. The elastic deformations *L*_e_ of the human–machine system are obtained: (6)$$L_{e}=\frac{F_{t}}{k_{\mathrm{total}}}$$

Hence, by combining Equations (5) and (6), the output displacement *L* of the traction unit can be acquired:(7)$$L=l_{AB}-l_{0}+\frac{F_{t}}{k_{\mathrm{total}}}$$

According to the profiles of *F*_t_ and *θ*, the desired displacement *L* can be calculated as shown in Figure 5. The displacement *L* reaches the maximum ${L^{\prime }}_{\max}=106\;\mathrm{mm}$ at *t* = *t*_90%_ and descends quickly to the initial position.

### 3.3. Assistance Strategy

When the hip angle is about to reach the minimum, the traction unit will be moved along the profile of the displacement *L* to produce the corresponding auxiliary force *F*_t_. Then the traction unit returns to the initial position immediately and waits for the next gait cycle. The drivers need to detect the current positions of the servo motors and consume the majority of the internal resource of the MCU. Besides, the algorithms of gait detection and prediction also occupy part of the internal resource. The motion lags appear in the movements of the traction unit and seriously restrict the performance of the H-Suit. Hence, the theoretical profile of the displacement *L* is simplified to a trapezoidal profile *L*_s_ with three segments, as shown in Figure 5. The traction unit moves from the initial position to the maximum with a constant speed, stays for a hold time *Δt* and returns to the initial position immediately.

According to the activation times of the iliopsoas muscle and the rectus femoris, the traction unit reaches the maximum position at *θ* = *θ*_zero_. The theoretical maximum ${L^{\prime }}_{\max}$ of the trapezoidal profile *L*_s_ can be calculated: (8)$${L^{\prime }}_{\max}={L^{\prime }}_{n}+{L^{\prime }}_{e}$$Where ${L^{\prime }}_{n}$ is the natural displacement *L*_n_ at *θ* = *θ*_max_/2 (*θ*_max_ denotes the maximum of hip angle *θ*) and ${L^{\prime }}_{e}$ is the elastic deformations *L*_e_ at *F*_t,max_ = 120 N. The deformation ${L^{\prime }}_{e}$ is acquired:(9)$${L^{\prime }}_{e}=\frac{F_{t,\max}}{k_{\mathrm{total},120}}$$
where *k*_total,120_ is the total stiffness *k*_total_ at *F*_t,ex_ = 120 N.

Due to the initial wearing deviations and the disturbances of the external force *F*_t_, the position deviations of the anchors are inevitable. A compensation displacement *L*_c_ is added to the theoretical maximum ${L^{\prime }}_{\max}$ for compensating the deviations. Hence, the maximum *L*_max_ of the trapezoidal profile *L*_s_ during the assistance process can be obtained:(10)$$L_{\max}={L^{\prime }}_{\max}+L_{c}$$

The compensation displacement *L*_c_ and the hold time *Δt* mainly affect the magnitude and duration of the peak force *F*_t, max_. With the help of the load cells, the auxiliary forces *F*_t_ during the gait cycle *T_i_* can be measured and the actual maximum $F_{t,\max}^{i}$ are selected. The displacement *L*_c_ can be obtained based on the forces $F_{t,\max}^{i}$ of the last *n* gait cycles.
(11)$$L_{c}=\frac{\sum_{i=1}^{n}\left(F_{t,\max}^{i}-F_{t,\max}^{}\right)}{n\cdot k_{\mathrm{total},120}}$$

The hold time *Δt* can improve the auxiliary profile and reduce the impacts of the forces *F*_t_. Under the constant peak forces, a long hold time *Δt* can enhance the assistance performance, especially the heartbeat index. There is a linear relationship between heart rate and oxygen consumption [34]. However, if the hold time *Δt* is too long, it will hinder the backward extension of lower limbs and aggravate the metabolic consumption. The traction and release times of the steel cable denoted by *Δt*_traction_ and *Δt*_release_ can be measured by the encoder of the control unit. The auxiliary time *ΔT* of the forces *F*_t_ can be calculated:*ΔT* = *Δt*_traction_+ *Δt*_release_+ *Δt*(12)

The auxiliary time *ΔT* is continuously changed for different walking speeds. Hence, during the assistance performance, the hold time *Δt* and the compensation displacement *L*_c_ are dynamically adjusted within the time *ΔT* according to the heartbeat index. Based on the control strategy, an algorithm is added to the control unit.

## 4. Performance Evaluation

### 4.1. Experimental Platform

There are three experiments for the performance evaluation: (I) Turn on H-Suit when walking with it (Powered); (II) Turn off H-Suit (Unpowered); (III) Not wear H-Suit (No-Suit). A subject performs each experiment by walking on the treadmill for 10 min and resting for 20 min, as shown in Figure 6. Eight subjects (average age 26, height 175 cm, weight 75 kg) wear the H-Suit and walk on the treadmill at the speed *v*_1_ = 0.75 m/s, as shown in Figure 7. When the curve of the hip angle *θ* is stable, the established prediction algorithm will be executed and the angle *θ* is predicted in real-time. Then, the traction units are opened and moved along the profile of the output displacements. The data of the angle *θ* and force *F*_t_ are measured by load cells and IMUs and recorded in the upper computer to analyze the assistance strategy.

### 4.2. Evaluation of Auxiliary Force

During the assistance experiments, the initial values of compensation displacement *L*_c_ and the hold time *Δt* are 17 mm and 300 ms, respectively. When a subject walked on the treadmill at the speed *v*_1_ = 0.75 m/s, the dynamic data of three gait cycles during the assistance process are shown in Figure 8. Compared with the real curve of *θ*, the prediction curve has a similar shape and advances at a time interval of 30 ms (Figure 8a). Meanwhile, the hip angle *θ* (such as the maximum, zero, minimum, etc.) can be forecasted early by a time of 30 ms. The predicted time of the minimum angle *θ*_min,_ which is the start time of the traction unit, will be sent to the control unit. The steel cables are pulled quickly to the maximum position (*L*_max_ = 123 mm) which is calculated by Equation (10), stayed for 300 ms and returned to the initial position immediately (Figure 8b). The traction of steel cables can produce the useful forces during the assistance phase where *F**^i^*_t,max_ = 135 N. Since the compensation displacement *L*_c_ is larger than the appropriate value, the actual peak forces of *F*_t_ are bigger than the ideal force *F*_t,max_ and has a slight impact on the hip joint. Similarly, the hold time *Δt* is also too long, the traction unit still stays at the maximum position even though the hip joint begins to stretch backward. When the H-Suit works without correctly adjusting the controller, the auxiliary forces *F*_t_ at the resistance phase hinder the extension movements at the later stage (Figure 8c). Hence, the improper values of *L*_c_ and *Δt* will interfere with the natural movements of lower limbs.

According to the effects of the two parameters, the established algorithm of the control strategy begins to work during performance experiments. Before the experiments, the steel cables should keep a tension state (*F*_t_ ≤ 5 N). A subject wears the H-Suit and walks on the treadmill at *v*_1_ = 0.75 m/s. Based on the peak forces *F^i^*_t,max_ and the average heart rates measured by a cardiotachometer, the optimum values of *L*_c_ and *Δt* are obtained through the control algorithm after 10 gait cycles where *L*_c_ = 7 mm and *Δt* = 200 ms. The changes in the average heart rates were analyzed and discussed in our previous work [31] where the heart rates have decreased remarkably. The corresponding dynamic data are shown in Figure 9. It can be found that the minimum *θ*_min_ is predicted 50 ms in advance and the traction unit starts to move along the profile of the displacements *L*. The traction unit can release the steel cable immediately after the hold time *Δt* and not hinder the deceleration of the flexion movement. Then the right leg completes the extension movements. The switching time from the flexion to extension is shorter and the real curve of *θ* is smoother compared with that of Figure 8a. Hence, the added mass and the traction movement of steel cables have few influences on the natural gait of lower limbs. The assistance phase dominates the whole activation interval of the H-Suit and the lag phase almost disappears. Although the traction unit tries to move along the ideal displacement curve with high precision, the real curve of *L* still has a few deviations at the peak stage. The real profile of the auxiliary forces *F*_t_ is in good agreement with the theoretical profile (Figure 9c) where *F^i^*_t,max_ is approximately equal to 120 N. The subject feels it is easier to walk under the assistance condition.

### 4.3. EMG Signals of Rectus Femoris and Vastus Lateralis

In order to evaluate the performance of H-Suit, the EMG signals of the rectus femoris and vastus lateralis of the subjects were measured through pasting electrodes according to SENIAM standards [35]. The corresponding EMG data were obtained at a normal speed (*v*_2_ = 1.25 m/s) and in three conditions (No-Suit, Unpowered, Powered), as shown in Figure 10. Figure 10a shows the EMG signals of the rectus femoris and vastus lateralis of 15 gait cycles under the condition of No-Suit. It can be found that the EMG signals of the two muscles have similar trends. While the periodic of the vastus lateralis is more obvious than the rectus femoris, the vastus lateralis was selected for further analysis. Figure 10b shows the EMG signals of the vastus lateralis of 5 gait cycles under the three conditions. The vastus lateralis has obvious periodic characteristics. Combined with the average rectified value (ARV) and median frequency (MNF) slopes in Figure 10c, it can be found that the change trends of the two indexes in the No-Suit and Unpowered are basically identical, while ARV decreases and MNF increases significantly at the powered condition. It shows that the H-Suit without the drive unit has little effect on the EMG signal of the lower limbs. In the powered condition, H-Suit can delay the muscle fatigue of the lower limbs where a steeper positive slope for ARV and a steeper negative one for MNF indicate a faster onset of fatigue [36]. The ARV slope during the powered condition was decreased by 0.5178 ± 0.1110/min and the MNF slope was increased by 0.2280 ± 0.0714/min.

In order to further evaluate the performance of the H-Suit, eight healthy subjects with a 10 kg load walked on the treadmill at three speeds (0.75 m/s, 1.25 m/s and 1.75 m/s) [37]. The electromyography (EMG) signals of the vastus lateralis were measured and analyzed, as shown in Figure 11. As the H-Suit provided the assistance for the flexion movements, it reduced the amount of effort that flexor muscle needed to exert. Figure 11a shows a representative case of the activation (raw EMG and its envelope) of the vastus lateralis during the three walking speeds, in both powered and unpowered conditions. The EMG’s values under the powered condition decreased markedly. Besides, the corresponding excitation was delayed and the action time was shortened. The net change in the vastus lateralis effort (Figure 11b), evaluated as the difference in the root mean square (RMS) of the EMG signals between the powered and unpowered cases, was significantly smaller than 0 for all velocities (*p* = 2 × 10^−3^, *p* = 7.0 × 10^−3^, *p* = 4 × 10^−3^ at *v* = 0.75 m/s, 1.25 m/s, 1.75 m/s). Figure 11c shows the change in the activation of the valtus lateralis as a percentage of its activation in the unpowered condition. The performance of the H-Suit degraded for higher speeds. Wearing the H-Suit resulted in a significant reduction of the vastus lateralis effort, averaged over subjects and walking speeds, of 13.3 ± 2.1% (*p* = 2 × 10^−5^).

## 5. Discussion

The ergonomic design of the H-Suit system is clearly presented, such as optimization of anchor points, comfort evaluation, analysis of biomechanical characteristics of lower limbs, etc. Compared with the development of the existed soft assistance systems [14,18,19], the design process is presented in more detail. The H-Suit processes the advantages of being lightweight, having high wearing comfort and has minimal impact on the natural gaits.

The profile of the auxiliary forces is planned in the auxiliary range where the forces start at the minimum of hip angle, reach the maximum (120 N) when the hip angle is equal to zero and end at 90% of each gait cycle. The beginning of the profile is dynamically changed according to the time of the minimum hip angle and the gait cycle. Compared with the profile proposed by Ding [22,23], our designed profile uses fewer memory resources and allows for faster processing speed. An assistance strategy is proposed to track the profile of the auxiliary forces by dynamically adjusting the compensation displacement *L*_c_ and the hold time *Δt*. The force-based position control [38] and admittance-based force tracking [39,40] are usually applied to track and compensate for the profile of auxiliary force. However, the assistance strategy proposed in this paper can achieve approximate performance and provide a new idea for the robotic suit. Furthermore, the influences of the variables *L*_c_ and *Δt* on the natural gaits and auxiliary forces have been analyzed to reveal the interactions between lower limbs and the H-Suit.

The changes in human metabolic consumption are essentially caused by the activation degree of muscles which can be evaluated by the slopes of the average rectified value and the median frequency [39]. During the performance evaluations, our H-Suit delayed the onset of fatigue and reduced the muscular activation of hip flexors. Wearing the H-Suit resulted in a significant reduction of the vastus lateralis effort, averaged over subjects and walking speeds, of 13.3 ± 2.1% (*p* = 2 × 10^−5^). A similar finding is described in [18], where a cable-driven suit for the hip extension is shown to reduce activation in muscles that do not cross the assisted joints (such as vastus lateralis). Unfortunately, a quantitative comparison here is not possible because of the different metrics used to assess fatigue in [18].

## 6. Conclusions

The advantages of a svelte and portable exosuit for the lower limbs, able to intuitively assist its wearer and reduce the effort required to walk, make it a good candidate for both industrial and clinical applications. Our results showed that the H-Suit can improve the walking endurance of lower limbs, delay muscle fatigue and reduce the activation level of hip flexors.

## 7. Limitations of the Study

The traction unit of the H-Suit is an external device and cannot be carried by the subjects. This study mainly analyzes the influences of the assistance strategy on the fatigue and activation of hip flexors. More flexors, especially the iliopsoas muscle, should be measured to evaluate metabolic consumption. Meanwhile, the experimental conditions (Powered, Unpowered and No-Suit) and walking speeds are not randomly distributed in the experiments of performance evaluation. In future research, a portable traction unit will be developed and subjects can complete all experiments when wearing the H-Suit system. The scheme of performance experiments will be improved by considering the randomization of experimental conditions.

## Figures and Tables

**Figure 1 micromachines-13-00825-f001:**
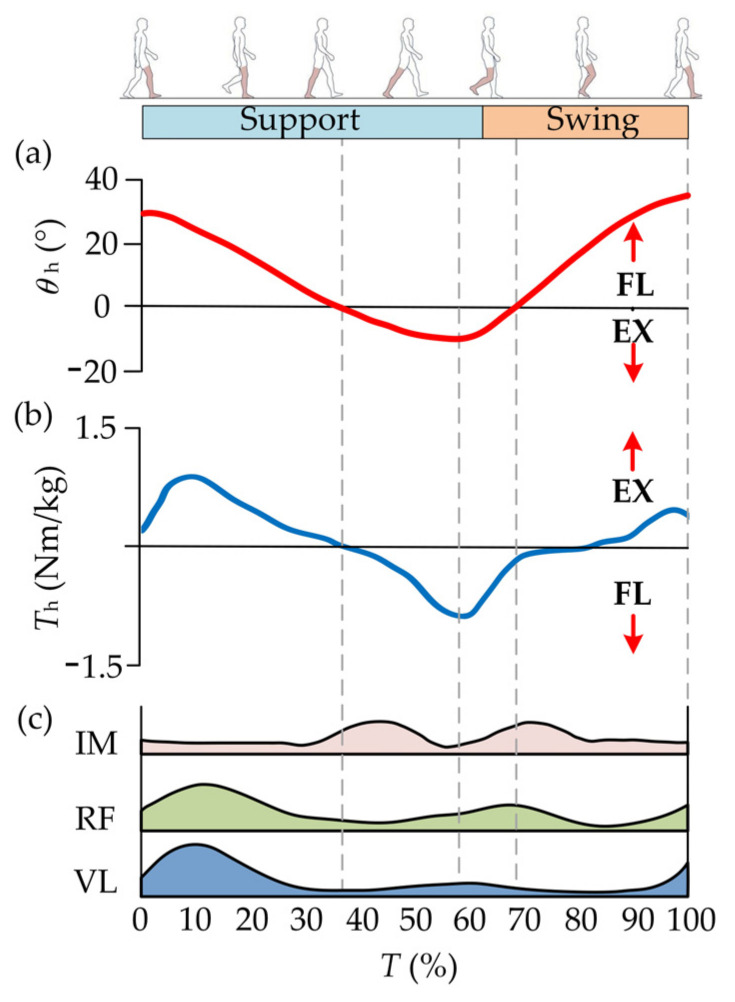
Kinematics and biomechanics of the hip joint. (**a**) When the hip joint moves to the absolute minimum angle (maximum extension), the thigh begins to flex forward and enter the early swing phase. (**b**) The flexion torque gradually increases with the forward flexion until the later stage of the swing phase. (**c**) The activations of the flexors are concentrated in the middle stage of the gait cycle [27].

**Figure 2 micromachines-13-00825-f002:**
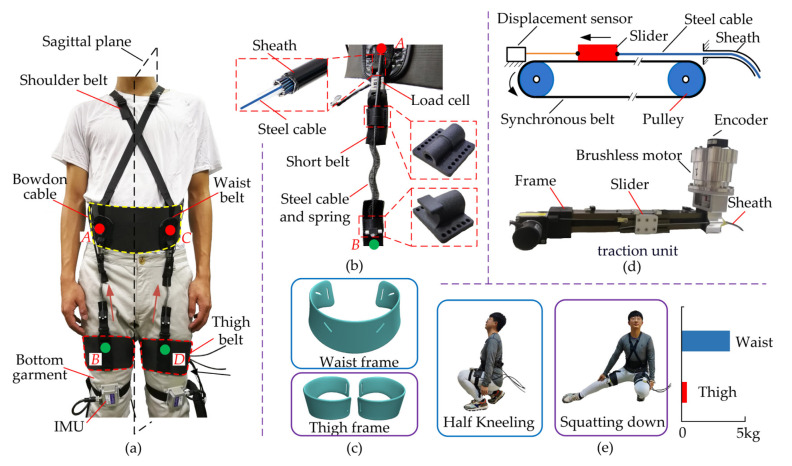
Development of the H-Suit system. (**a**) The H-Suit consists of a soft exosuit, external traction units and Bowden cables where *A*, *B*, *C* and *D* are anchor points. (**b**) The sheath and steel cable of the Bowden cable are respectively fixed in the upper and lower components. (**c**) The open frames of the waist and thigh are designed and fitted with the physiological structures of the human body. (**d**) The synchronous belt transmission is adopted in the traction unit and the steel cable is reciprocally dragged by the slider. (**e**) The H-Suit has few constraints on the dexterity of low limbs and daily activities.

**Figure 3 micromachines-13-00825-f003:**
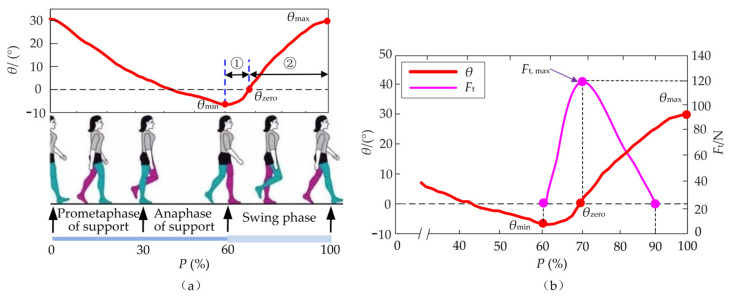
Profile of auxiliary forces *F*_t_. (**a**) The hip flexion is divided into two stages: ① The first stage is from *θ*_min_ to the zero-point *θ*_zero_, ② The second one is from *θ*_zero_ to *θ*_max_. (**b**) The profile of the auxiliary force *F*_t_ begins at *t* = *t_θ_*_,min_, reaches the peak value *F*_t, max_ at *t* = *t_θ_*_,zero_, and ends at *t* = *t*_90%_.

**Figure 4 micromachines-13-00825-f004:**
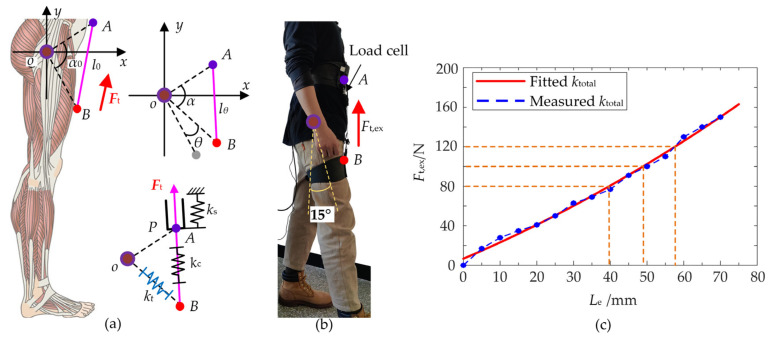
Measurement of the total stiffness. (**a**) The desired displacement consists of the natural displacement of Bowden cables and the elastic deformations of the human–machine system. (**b**) The total stiffness *k*_total_ was measured when the hip flexed forwards 15°. (**c**) The external force *F*_t_,_ex_ at the equal intervals of displacements are presented and *k*_total_ can be fitted.

**Figure 5 micromachines-13-00825-f005:**
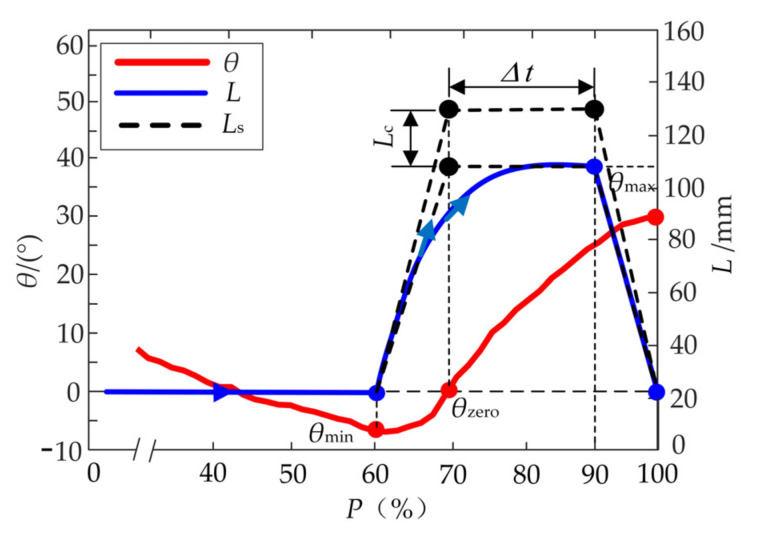
Desired displacement *L*.

**Figure 6 micromachines-13-00825-f006:**
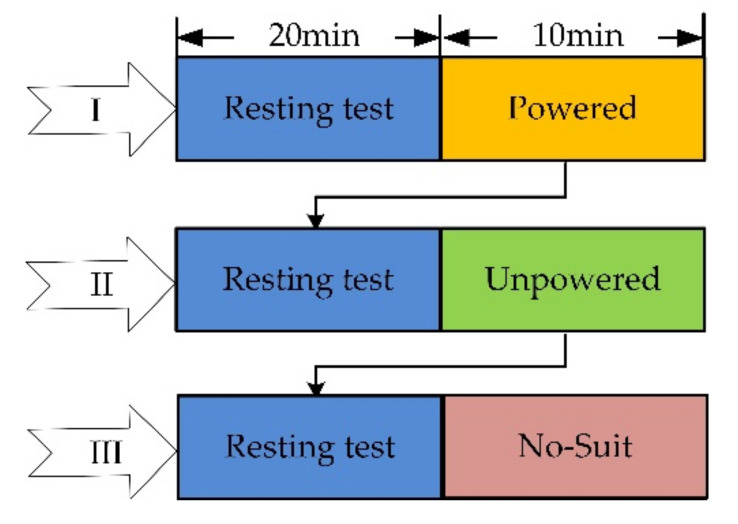
Scheme of assistance experiments. The subjects rested for 20 min before the test and walked in the state of Powered, Unpowered and No-Suit for 10 min.

**Figure 7 micromachines-13-00825-f007:**
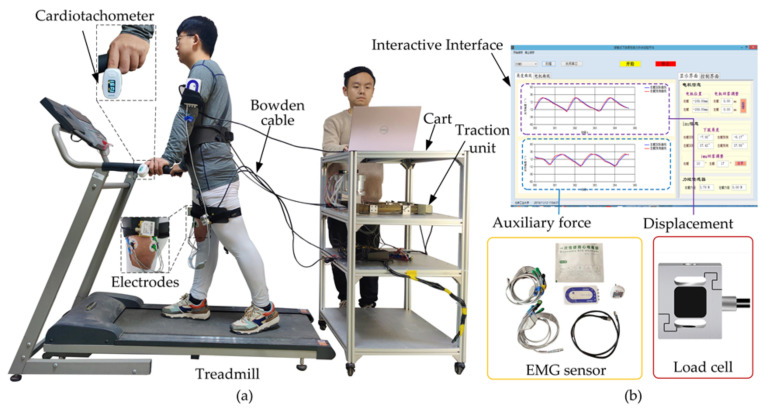
Experimental platform of the H-Suit. (**a**) The subjects walked on the treadmill at three different speeds. (**b**) The data of EMG, auxiliary forces and displacements are measured and depicted in the visual interface.

**Figure 8 micromachines-13-00825-f008:**
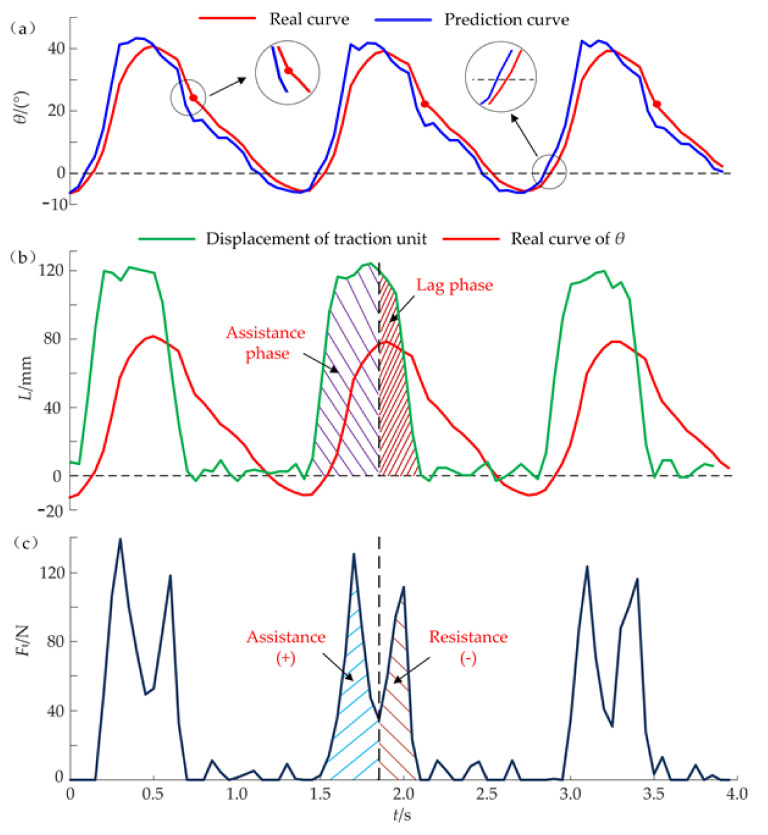
Dynamic data during the assistance process (*v*_1_ = 0.75 m/s, *L*_c_ = 17 mm, *Δt* = 300 ms). (**a**) The prediction curve of the hip angle *θ* has a similar shape and advances a time interval compared with the real curve. (**b**) The output displacement *L* is determined by the parameters *L*_c_ and *Δt*. (**c**) The auxiliary forces *F*_t_ hinder the extension movements at the later stage due to the improper values of *L*_c_ and *Δt*.

**Figure 9 micromachines-13-00825-f009:**
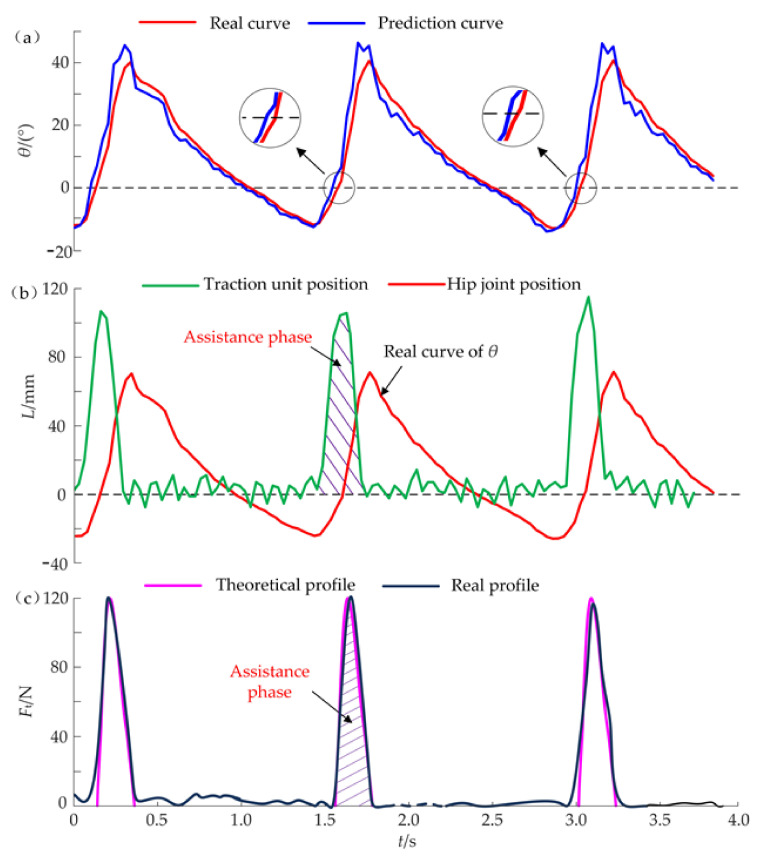
Experimental data of the effective assistance (*v*_1_ = 0.75 m/s, *L*_c_ = 7 mm, *Δt* = 200 ms) (**a**) The switching time from the flexion to extension is shorter and the real curve of *θ* is smoother compared with that of Figure 8a. (**b**) The assistance phase dominates the whole activation interval of the H-Suit and the lag phase almost disappears. (**c**) The real profile of the auxiliary forces *F*_t_ is in good agreement with the theoretical profile.

**Figure 10 micromachines-13-00825-f010:**
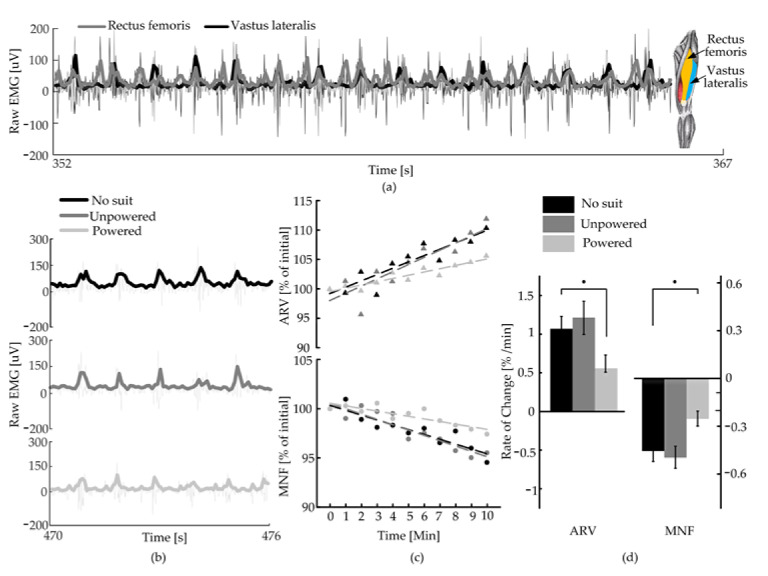
Fatigue analysis. (**a**) Raw signal and envelope of EMG of the rectus femoris and vastus lateralis of one subject at a normal speed (*v* = 1.25 m/s). (**b**) Raw signal and envelope of EMG of the vastus lateralis in the conditions of No-Suit, Unpowered and Powered. (**c**) Trend of the average rectified value (ARV) and median frequency (MNF) of the EMG signal of one subject in the 10 min of tests. Indexes are expressed in percentage of their initial value. (**d**) Slope of ARV and MNF. Both indexes confirm that wearing the H-Suit significantly reduces the onset of fatigue (*p* = 0.03 for ARV and *p* = 0.01 for the MNF). Error bars show the standard error of the mean.

**Figure 11 micromachines-13-00825-f011:**
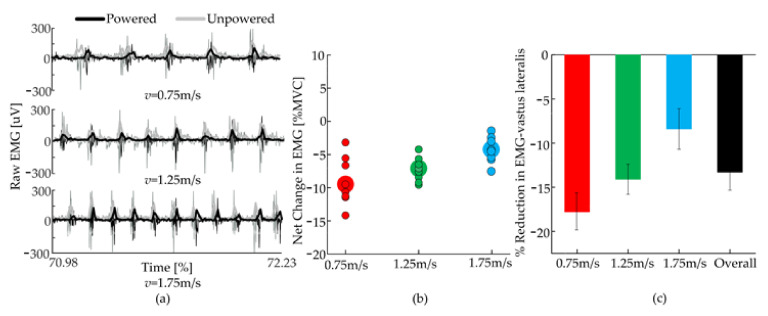
Changes in muscular activation. (**a**) Raw signal and envelope of the vastus lateralis’ EMG. (**b**) Net change (Powered—Unpowered) of RMS of the EMG signals of the vastus lateralis, for the three walking speeds. Opaque contoured circles are the values for each individual subject, bigger circles indicate the mean over subjects. (**c**) Changes in the activation of the vastus lateralis, expressed as a percentage of its activation in the unpowered condition (net change/Unpowered). Error bars show the standard error of the mean.

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
