# Peer review of "Ergonomic Design and Performance Evaluation of H-Suit for Human Walking"

_micromachines, 2022, doi:10.3390/mi13060825_

Round 1

Reviewer 1 Report

I have added my comments to the PDF, so I provide here three general comments:

You must improve the quality of English, consider having a native English speaker proof reading the text.

There is a clear lack of a "limitations of the study" section, where you should state the lack of randomization in the design of the experiments.

You should add a discussion section, to discuss and compare to the current state of the art your findings.

Author Response

Thank you for your valuable comments, please see the attachment.

Reviewer 2 Report

In this paper, Zhang et al reported an ergonomic desgin and performance evalutation of a lower body device for human walking. Overall, this paper is well organized and well written. The reviewer can recommend for publication with the following comments:

  1. The novelty is not clearly introduced in the begining of this paper. In the introduction section, the authors listed a series of other researchers' works, however, the limitations of these works, hence the novelties of this work, are not mentioned clearly.
  2. The labels ((a),(b)...) in different figures are in different fonts and some of them are in bold while some are not. Please unify the labels in all figures.
  3. Figure 2 is too crowded, which makes it difficult to read. The different sub-figures should be separated more.

Author Response

(The authors gave the same response as above.)

Round 2

Reviewer 1 Report

Please, take again last revision document and go through all the comments. I encourage you to pay attention to potential situations where a external reader could find concepts, that may be clear for you who know the full content of the investigation, totally unclear or new.

Author Response

Thank you very much for your valuable comments. I have gone through all the comments proposed at the first round. Some inappropriate descriptions and grammatical errors were corrected and marked in the last revision of this manuscript. The potential situations, such as application area, assistance performance and robotic suit, have been presented and partially modified in this manuscript. 
The detailed revisions are marked in the second revision document.
